# Performance Analysis of Time Series Deep Learning Models for Climate Prediction in Indoor Hydroponic Greenhouses at Different Time Intervals

**DOI:** 10.3390/plants12122316

**Published:** 2023-06-14

**Authors:** Oybek Eraliev, Chul-Hee Lee

**Affiliations:** 1Department of Future Vehicle Engineering, Inha University, 100 Inharo, Mitchuholgu, Incheon 22212, Republic of Korea; oybekeraliev7@gmail.com; 2Department of Mechanical Engineering, Inha University, 100 Inharo, Mitchuholgu, Incheon 22212, Republic of Korea

**Keywords:** time series, hydroponic greenhouse, climate prediction, convolutional neural network, deep neural network, long–short-term memory

## Abstract

Indoor hydroponic greenhouses are becoming increasingly popular for sustainable food production. On the other hand, precise control of the climate conditions inside these greenhouses is crucial for the success of the crops. Time series deep learning models are adequate for climate predictions in indoor hydroponic greenhouses, but a comparative analysis of these models at different time intervals is needed. This study evaluated the performance of three commonly used deep learning models for climate prediction in an indoor hydroponic greenhouse: Deep Neural Network, Long–Short Term Memory (LSTM), and 1D Convolutional Neural Network. The performance of these models was compared at four time intervals (1, 5, 10, and 15 min) using a dataset collected over a week at one-minute intervals. The experimental results showed that all three models perform well in predicting the temperature, humidity, and CO_2_ concentration in a greenhouse. The performance of the models varied at different time intervals, with the LSTM model outperforming the other models at shorter time intervals. Increasing the time interval from 1 to 15 min adversely affected the performance of the models. This study provides insights into the effectiveness of time series deep learning models for climate predictions in indoor hydroponic greenhouses. The results highlight the importance of choosing the appropriate time interval for accurate predictions. These findings can guide the design of intelligent control systems for indoor hydroponic greenhouses and contribute to the advancement of sustainable food production.

## 1. Introduction

Indoor hydroponic greenhouses provide an efficient and sustainable method of food production in urban areas where arable land is available. In hydroponic systems, plants are grown in a nutrient-rich solution instead of soil, which allows for precise control of the growing conditions. On the other hand, maintaining optimal growing conditions inside the greenhouse is crucial for achieving high crop yields [1]. The climate conditions inside the greenhouse, including temperature, humidity, and CO_2_ concentration, need to be monitored and controlled to ensure optimal plant growth and development. Climate prediction in indoor hydroponic greenhouses is complex due to the interdependence of different variables and their nonlinear relationships. Traditional methods for climate predictions, such as statistical models, have limitations in capturing the complex patterns and relationships between different variables [2,3,4,5]. Time series deep learning models have the potential for analyzing sequential data and making accurate predictions. These models can learn from the historical climate data and identify patterns and trends that are difficult to detect using traditional methods [6].

This study compared the performance of three commonly used time series deep learning models to predict the climate conditions in an indoor hydroponic greenhouse at different time intervals: Deep Neural Network (DNN), Long–Short Term Memory (LSTM), and 1D Convolutional Neural Network (1D-CNN). The dataset used in this study was collected over a week at one-minute intervals and prepared at four different intervals: 1, 5, 10, and 15 min. The performance of the models was evaluated based on several metrics, including Mean Absolute Error (MAE), Root Mean Square Error (RMSE), the percent standard error of the prediction (% SEP), and Coefficient of Determination (R^2^) [1].

The key findings of the study are as follows:the LSTM model outperformed the other models in all time intervals in predicting the temperature and humidity, achieving the lowest MAE, RMSE, SEP, and the highest R-squared values;the increase in the time interval adversely affects the performance of the models;the DNN model performed better than the 1D-CNN model but not as well as the LSTM model;the performance of the models varied for different climate variables, with temperature being the easiest to predict and humidity being the most challenging.

These results provide insights into the effectiveness of time series deep learning models for climate prediction in indoor hydroponic greenhouses. The LSTM model outperformed the other models at shorter time intervals, whereas the DNN model showed the best performance at longer time intervals. Increasing the time interval from 1 to 15 min adversely affected the performance of the models. These findings can guide the development of intelligent control systems for indoor hydroponic greenhouses and contribute to the advancement of sustainable food production. Nevertheless, the study had some limitations. First, the dataset used in the study was limited to one week, which may not represent the long-term climate conditions in a hydroponic greenhouse. Second, the study only considered three deep-learning models. The other models may have better performance for climate prediction. Finally, the study did not consider external factors, such as weather conditions and plant growth, which might affect the climate conditions in the greenhouse.

## 2. Related Work

In recent years, there has been growing interest in applying time series deep learning models for climate predictions in various domains, including indoor hydroponic greenhouses [7,8]. These models can potentially capture the complex relationships between different variables and make accurate predictions based on historical data. This section reviews the existing literature on time series deep learning models for climate prediction and identifies the gaps the current study aims to fill.

Deep Neural Networks (DNN) are a type of deep learning model that can learn complex nonlinear relationships between input and output data. DNN is used widely for climate predictions in various domains. DNN models are useful for predicting nonlinear systems because they can model such systems without making assumptions implicit in most traditional statistical approaches [9,10]. DNN models have several advantages over other nonlinear models because they can approximate a broad class of functions with high accuracy, making them universal approximators. These models have been used to forecast greenhouse climatic data with superior results to physical models [11,12,13]. On the other hand, there are some limitations to using DNN models, such as optimization issues, applicability to real-world problems, over-fitting, the need for many training sets, and poor stability in strongly coupled and complex systems [12]. Several studies have proposed time-series models to provide reduced representations of large numerical systems for an accurate simulation and prediction of the dynamic responses. Since Lapedes and Farber [14] combined a nonlinear time series model with DNN, this approach has attracted attention for integrating machine learning algorithms and regression models through methods, such as autoregressive moving average model (ARMAX), nonlinear autoregressive network (NARX), and autoregressive integrated moving average model (ARIMA). NARX is a class of dynamic DNN models applied widely in various fields because they can represent any nonlinear function based on the ARX input [15].

LSTM is a recurrent neural network (RNN) that can handle sequential data with long-term dependencies. LSTM is used widely for time series analysis and prediction in various domains. In climate predictions, LSTM has been applied to predict different variables, including temperature, humidity, and wind speed. Liu et al. [6] used an LSTM model to predict the temperature and humidity in a greenhouse. They compared the performance of the LSTM and RNN models and reported that the LSTM model outperformed the RNN model. Hu et al. [16] proposed a hybrid model that combined LSTM with a differential evolution algorithm for predicting wind speed. They reported improved accuracy compared to traditional statistical models.

CNN is a type of deep learning model that can learn spatial and temporal patterns in data. CNN is used widely for image and signal processing and shows promising results for time series analysis and prediction. In climate predictions, CNN has been applied to predict different variables, including temperature, humidity, and CO_2_ concentration. For example, Jin et al. [17] applied a CNN and LSTM model to predict temperature, humidity, and wind speed. They reported that the CNN model showed better performance than traditional statistical models. Tzoumpas et al. [18] proposed a data-filling methodology and used CNN and LSTM models to predict indoor temperature. They reported improved accuracy compared to traditional statistical models.

Despite the promising results, few studies have compared the performance of different deep-learning models for climate prediction in indoor hydroponic greenhouses. Most studies focused on predicting a single variable, such as temperature or humidity, while in hydroponic systems, multiple variables need to be predicted and controlled simultaneously. Therefore, this study aims to fill the gap in the literature by evaluating the performance of three commonly used deep learning models, i.e., DNN, LSTM, and 1D-CNN, to predict multiple climate variables in an indoor hydroponic greenhouse at different time intervals.

## 3. Results and Discussion

Table 1 lists the results of the accuracy assessment of the DNN model. The temperature showed the highest level of precision among all parameters. After one minute of DNN-1, the temperature prediction showed an MAE of 0.005, an RMSE of 0.006, an SEP of 0.6%, and an R^2^ of 0.98. In contrast, the predictions for humidity and CO_2_ were relatively less accurate, with MAEs of 0.02 and 0.01, RMSEs of 0.03 and 0.01, SEPs of 3.04% and 3.45%, and R^2^ values of 0.96 and 0.81, respectively. DNN has performed its best result with R^2^ values of 0.99, 0.95, and 0.97, SEPs of 0.4, 3.65, and 1.08 for temperature, humidity, and CO_2_ level, respectively when the time interval is five 5 min. Furthermore, the DNN-15 predictions had a SEP of 0.56% (temperature), 4.22% (humidity), and 2.15% (CO_2_), which was superior to the other models.

Table 2 and Table 3 list the findings of the LSTM and 1D-CNN models, respectively. The overall performance of the 1D-CNN model was inferior to LSTMs and DNN, but it was slightly better in predicting CO_2_ than the DNN and LSTM models when the time interval was one minute. For example, 1D-CNN-1 accurately predicted the temperature, humidity, and CO_2_ with an R^2^ of 0.96, 0.96, and 0.97, respectively. On the other hand, the 1D-CNN model accuracy decreased as the prediction time went beyond five minutes and showed the lowest result in predicting temperature and CO_2_ when the time interval was 15 min. On the other hand, the predictive accuracy of the LSTM model was superior to all other models. LSTM-5 correctly predicted the temperature with an MAE of 0.004, RMSE of 0.005, SEP of 0.46%, and R^2^ of 0.99.

Figure 1, Figure 2 and Figure 3 compare the model performance through bar charts. These charts depict the prediction error of each model in terms of SEP and illustrate the changes in prediction error at each time step. Figure 1 shows the SEP variation for the predicted temperature. While the accuracy of DNN fluctuated over time, the accuracy of 1D-CNN and LSTM decreased with time. For example, 1D-CNN showed superior performance at the 10 min time step; it was much less accurate than DNN and LSTM in the remaining time steps. For the humidity prediction model, the DNN and LSTM time-based algorithms showed an increase in error with time, while 1D-CNN showed unstable performance, as shown in Figure 2. The humidity prediction was more challenging for all time-series-based models. On the other hand, the LSTM model showed relatively better results than DNN and 1D-CNN, as shown in Figure 3. Increasing the time interval adversely affected the prediction of the environment of the hydroponic greenhouse of all DL models used.

Figure 4, Figure 5 and Figure 6 show the performance of the DNN, 1D-CNN, and LSTM models for predicting temperature, humidity, and CO_2_ level under varying time steps. The accuracy of all three models was relatively high in predicting the temperature across the three climate conditions. The figures show the prediction accuracy of temperature, humidity, and CO_2_ for each model.

This study shows the potential of utilizing data-based modeling techniques to forecast environmental alterations in greenhouses. The LSTM model incorporating a time series-based algorithm performed better than the conventional neural network-based models. Therefore, employing a more dynamic modeling approach incorporating previous experiences is advantageous for predicting continuous and repetitive changes in the special greenhouse environment. Previous studies [19,20] reported similar findings.

These findings highlight the importance of selecting the appropriate time series deep learning model for climate prediction in indoor hydroponic greenhouses. The LSTM model was the best choice for accurate time-interval predictions, while DNN and 1D-CNN models showed relatively low accuracy. This study highlights the significance of including more data in the training dataset to improve the performance of the models. The strengths of this study include the use of multiple deep-learning models, different time intervals, and multiple performance metrics for evaluation. The study also provides valuable insights for researchers and practitioners in hydroponics and agriculture interested in climate prediction using deep learning models. Nevertheless, this study also had some limitations. First, the dataset used in the study is limited to one week, which may not represent the long-term climate conditions in a hydroponic greenhouse. Second, the study only considered three deep learning models; other models may perform better for climate predictions. Finally, the study did not consider external factors, such as weather conditions and plant growth, which can affect the climate conditions in the greenhouse. In conclusion, this study examined climate predictions in indoor hydroponic greenhouses by evaluating the performance of different time series deep learning models. The study provides insights into the strengths and weaknesses of the models and highlights the importance of selecting appropriate models for accurate predictions. Further research will consider external factors and use larger datasets to improve the performance of the models.

## 4. Materials and Methods

### 4.1. Greenhouse Measurements and Dataset

The experiment was conducted in an indoor hydroponic greenhouse depicted in Figure 7a with an experimental area of 50 × 50 × 60 cm. A fully automated indoor hydroponic greenhouse was designed, which involved various stages, such as plant and disease detection, environment prediction, AI-based decision-making, and automation. This study focused on the environmental prediction aspects of the original investigation. The predicted variables can be utilized in a control scheme to dynamically adjust and regulate various environmental factors within the greenhouse. For instance, based on the predicted temperature, the control system can activate or deactivate heating or cooling systems to maintain the optimal temperature range for crop growth. Similarly, the predicted humidity levels can guide the activation of humidifiers or dehumidifiers to ensure the appropriate moisture content in the air. Additionally, the predicted CO_2_ levels can be used to regulate ventilation systems and control the inflow and outflow of air to maintain the desired CO_2_ concentration. Integrating the predicted variables into a control scheme creates an automated and optimized environment for crop production. This directly impacts crop growth, as optimal climate conditions are vital for achieving higher yields, reducing disease incidence, and improving overall crop quality. Precise control of temperature, humidity, and CO_2_ levels can influence various physiological processes in plants, including photosynthesis, transpiration, and nutrient uptake. Maintaining these variables within the optimal range can promote healthy plant growth, improve resource utilization, and ultimately enhance crop productivity. Figure 7b presents the condition monitoring system for the hydroponic greenhouse. In the greenhouse, controlling fans, lights, and pumps are an essential aspect of creating an optimal environment for crop growth. These components are typically controlled through an automated system that considers various factors, including the predicted values of climate variables. The control strategy for managing fans, lights, and pumps can be based on a combination of real-time sensor data and the predicted values of temperature, humidity, and CO_2_ levels. The goal is to maintain the desired setpoints for these variables and ensure their stability within the optimal range. For example, fans can be controlled to regulate air circulation and maintain proper ventilation within the greenhouse. Based on the predicted temperature and humidity levels, the control system can adjust the fan speed or activate/deactivate specific fans to achieve the desired airflow and prevent overheating or excessive moisture buildup. Similarly, the control of lights in the greenhouse can be guided by the predicted values of climate variables. The lighting duration and intensity can be adjusted to provide the optimal light spectrum and duration for different stages of plant growth. By considering the predicted values, the control system can ensure that the lighting conditions are tailored to the specific requirements of the crops. Pumps are used in hydroponic systems to deliver nutrient solutions to the plants. The control of pumps can be synchronized with the predicted climate variables to optimize the delivery of nutrients. For instance, based on the predicted water consumption by the plants (indicated by the humidity levels), the control system can adjust the pump operation to deliver the appropriate amount of nutrient solution, ensuring efficient water and nutrient management. The greenhouse consists of four types of plants, i.e., lattice, kale, chicory, and spinach, and is equipped with a camera (for plant detection as part of this investigation) and a sensor module (Model: SCD40, Figure 7a). All the components are controlled by Jetson Nano, and the DL models are also trained on the Jetson Nano.

The dataset used in this study was collected from an indoor hydroponic greenhouse for one week (16 March to 23 March 2023), where the temperature, humidity, and CO_2_ levels were recorded every minute. The dataset was collected using sensors placed inside the greenhouse connected to a data acquisition system. The dataset was divided into four different time intervals, i.e., 1, 5, 10, and 15 min. The dataset was preprocessed to remove any missing values and outliers. The preprocessing also includes the normalization of the data to make the range of data between 0 and 1. The data was then divided into training and testing datasets, with 80% of the data used for training and the remaining 20% for testing. Feature engineering was performed on the dataset to extract the relevant features for the deep learning models. The extracted features included the lagged temperature, humidity, and CO_2_ levels. These features were used as input to the deep learning models.

### 4.2. DL Models for Forecasting Environmental Changes

Three different deep-learning models are used in this study for climate prediction in indoor hydroponic greenhouses. The models are as follows:

A) The DNN model is a feedforward neural network that takes the input features and passes them through a series of fully connected layers. The output layer produces the predicted temperature, humidity, and CO_2_ levels. The input nodes transfer their signals to the nodes in the three hidden layers. The distribution of these signal values depends on the connection weights between the input and hidden nodes. The hyperbolic tangent function (tanh) and the rectified linear unit (ReLu) function are transferred functions for the hidden layers as expressed in Equations (1) and (2), respectively.
(1)$$f\left(x\right)=\frac{2}{1+e^{-2x}}-1$$
(2)$$f\left(x\right)=\left\{\begin{array}{c} 0,\;for\;x<0\\ x,\;for\;x\geq 0 \end{array}\right.$$

The Adam optimization algorithm is applied during training to accelerate learning and improve convergence. The training stops repeating when the cost function calculated using the predicted value decreases below a specific value or when the number of iterations reaches the set number of times. The three models used in this study were trained with 50 iterations and a cost value of 0.00013. The hidden layers of the DNN model consisted of 128, 64, and 32 hidden nodes that update their weights through the Adam optimizer with a learning rate of 0.001. This optimal condition is determined by trial and error. Figure 8 presents the architecture and schematic representation of the neural network used in this study. The DNN model was built using the TensorFlow 2.0 open library based on Python 3.9.

B) The LSTM model is a type of recurrent neural network that can handle long-term dependencies in the data. The LSTM model takes the input features and passes them through a series of LSTM layers. The output layer produces the predicted temperature, humidity, and CO_2_ levels. The LSTM structure is constructed using the time series of multivariate inputs and passing them through the LSTM layer, and the number of outputs is equivalent to the number of neurons, as shown in Figure 9a. The LSTM layer consists of 64 cells. The LSTM model was built using the TensorFlow 2.0 and Keras libraries based on Python 3.9.

The LSTM network comprises cells that use the input from the previous state and current input to predict the next output. The cells are responsible for determining the importance of the data, storing it in memory, and transferring it to the next loop or ignoring it, allowing the RNN to overcome the vanishing gradient in long-time series analysis. Figure 9b shows the LSTM cell structure that operates in the following order: the forget gate (Equation (3)) receives information $h_{t-1}$ and $X_{t}$, outputs a number between 0 and 1 for each number in the cell state $C_{t-1}$, where an output number of 1 retains the data completely while 0 deletes the data.
(3)$$f_{t}=\sigma (W_{f}\cdot \left[h_{t-1},X_{t}\right]+b_{f})$$

The following stage is split into two parts: updating the cell state and activating the input gate, corresponding to Equations (4) and (5). To determine the new cell state, the old cell state is multiplied by the forget gate and added to the input gate multiplied by the updated cell state, as shown in Equation (6).
(4)$$i_{t}=\sigma (W_{i}\cdot \left[h_{t-1},X_{t}\right]+b_{i})$$
(5)$${\hat{C}}_{t}=tanh(W_{C}\cdot \left[h_{t-1},X_{t}\right]+b_{C})$$
(6)$$C_{t}=f_{t}*C_{t-1}+i_{t}*{\hat{C}}_{t}$$

The sigmoid layer of the output gate determines the cell state (according to Equation (7)). The result is then passed through the tanh function and multiplied by the output of the sigmoid gate, which is obtained using Equation (8).
(7)$$O_{t}=\sigma (W_{O}\cdot \left[h_{t-1},X_{t}\right]+b_{O})$$
(8)$$h_{t}=O_{t}*tanh\;(C_{t})$$

C) The 1D-CNN model is a convolutional neural network that takes the input features and passes them through convolutional layers. The output layer produces the predicted temperature, humidity, and CO_2_ levels. First, the input values pass through two convolutional layers. The convolutional layers apply 64 filters to the input data, which is one-dimensional. The filters slide over the input data, performing a dot product with the values in the input at each position. The results of this dot product are called the convolutional output. The output of each filter is then passed through an activation function (ReLU), which adds non-linearity to the model. This process is repeated for each filter, resulting in an output tensor with several channels equal to the number of filters. The output tensor from the convolutional layers is then passed through a maximum pooling layer. The maximum pooling layer downsamples the output tensor by taking the maximum value from each non-overlapping subregion of the tensor. This helps reduce the spatial dimensionality of the data and makes the network more efficient. After the maximum pooling layer, the output is flattened and passed through two dense layers with 32 and 16 nodes, respectively. The dense layers perform a matrix multiplication on the flattened output of the previous layer and a set of learned weights. Each neuron in the dense layer takes a linear combination of the previous layer’s outputs as the input and outputs a scalar value. These scalar values are passed through another activation function (ReLU), which introduces non-linearity into the model. The final output layer has three nodes, which output the final prediction for the network, as shown in Figure 10.

### 4.3. Performance Metrics

The correlation between the predicted and observed data was evaluated using the coefficient of determination (R^2^) to compare the prediction models in this study. Furthermore, the accuracy of the model was assessed by calculating the percent standard error of the prediction (% SEP), the root mean square error of prediction (RMSE), and the mean absolute error (MAE). These metrics help to determine how well the model explains the differences between predicted and observed values [12]. The formulae for these calculations are given in Equations (9)–(12),
(9)$$MAE=\frac{\sum_{i=1}^{N}\left|y_{i}-x_{i}\right|}{N}$$
(10)$$RMSE=\sqrt{\frac{\sum_{i=1}^{N}{\left(x_{i}-y_{i}\right)}^{2}}{N}}$$
(11)$$R^{2}=1-\frac{SSE}{SSTO}=1-\frac{\sum_{i=1}^{N}{\left(x_{i}-y_{i}\right)}^{2}}{\sum_{i=1}^{N}{\left(x_{i}-{\bar{x}}_{i}\right)}^{2}}$$
(12)$$SEP(\%)=\frac{100}{{\bar{x}}_{i}}\sqrt{\frac{\sum_{i=1}^{N}{\left(x_{i}-y_{i}\right)}^{2}}{N}}$$
where the *SSTO* measures the variability of mean observed values; *n* is the total number of data sets used for estimation; $x_{i}$ is the actual temperature (observed output); $y_{i}$ is the predicted temperature (estimated output); ${\bar{x}}_{i}$ is the mean value of the observed outputs of the prediction set. Ideally, a perfect match between the predicted and observed values would result in an R^2^ value close to 1 and a (%) SEP value close to 0.

## 5. Conclusions

This study evaluated the performance of three different time series deep learning models for climate predictions in indoor hydroponic greenhouses. The LSTM model outperformed the other models in all time intervals, achieving the lowest MAE, RMSE, and SEP (%) and the highest R-squared values. The models’ performance decreased as the dataset’s time interval increased. The main contributions of this study include evaluating multiple deep-learning models, different time intervals, and multiple performance metrics for predicting the climate in indoor hydroponic greenhouses. The study provided insights into the strengths and weaknesses of the models and highlighted the importance of selecting appropriate models for accurate predictions. The implications of this study for indoor hydroponic greenhouse climate prediction were significant. Accurate climate predictions can help optimize the growth of plants, reduce energy consumption, and increase the yield and quality of crops. The study guides researchers and practitioners in hydroponics and agriculture interested in using deep-learning models for climate prediction in indoor hydroponic greenhouses. This study demonstrated the effectiveness of deep learning models for climate prediction in indoor hydroponic greenhouses. The study provides insights into selecting appropriate models for accurate predictions, and its implications for hydroponics and agriculture are significant. The findings of this study can guide future research in this area and help optimize plant growth, reduce energy consumption, and increase the yield and quality of crops.

Future research directions can extend this study by considering external factors, such as weather conditions and plant growth, which may affect the climate conditions in the greenhouse. The study can also be extended using larger datasets, longer time intervals, and more advanced deep-learning models. Furthermore, research can explore the transferability of the models to other indoor farming systems and assess the feasibility of implementing the models in real-time greenhouse climate prediction systems.

## Figures and Tables

**Figure 1 plants-12-02316-f001:**
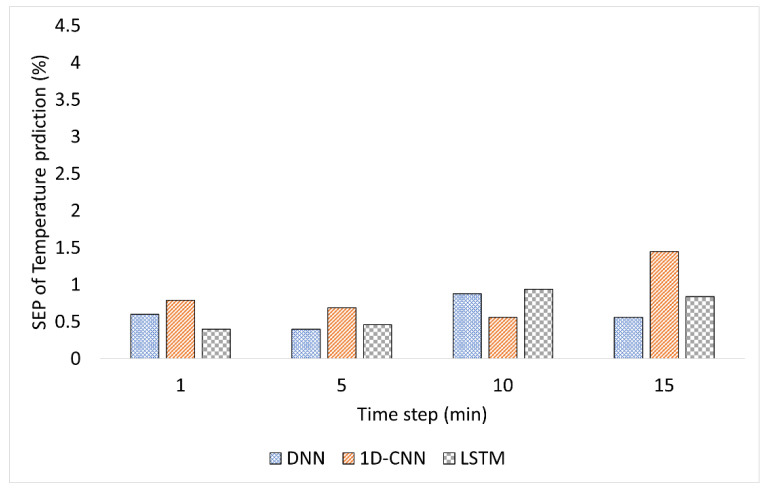
Comparison of the SEP per time for a temperature prediction.

**Figure 2 plants-12-02316-f002:**
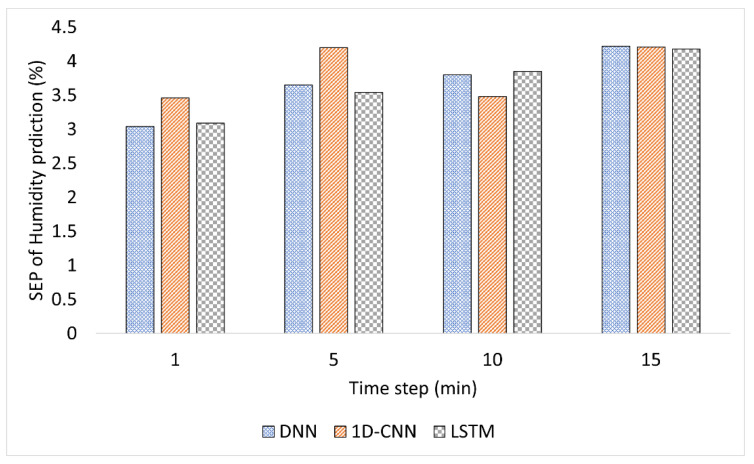
Comparison of the SEP per time for humidity prediction.

**Figure 3 plants-12-02316-f003:**
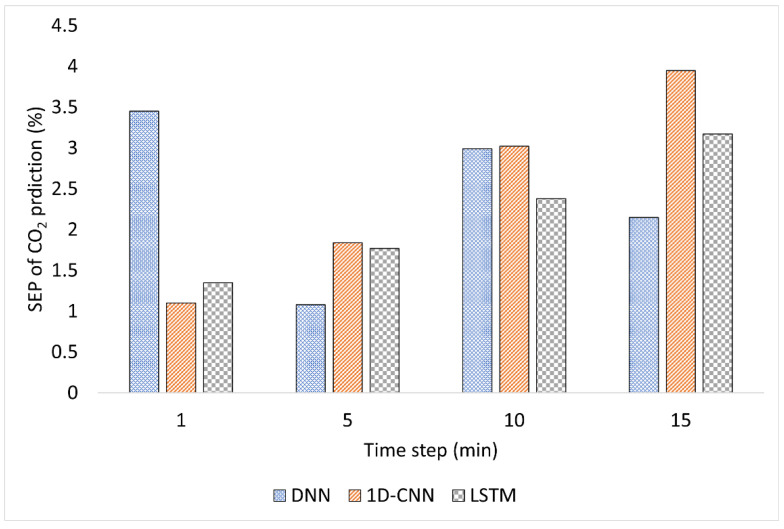
Comparison of the SEP per time for CO_2_ prediction.

**Figure 4 plants-12-02316-f004:**
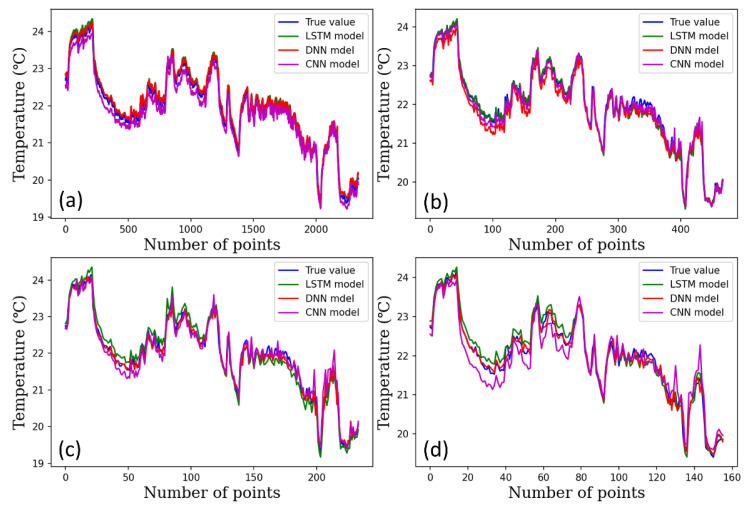
Temperature prediction results using the DL models per time steps, (**a**) 1-min time step, (**b**) 5-min time steps, (**c**) 10-min time steps, (**d**) 15-min time steps.

**Figure 5 plants-12-02316-f005:**
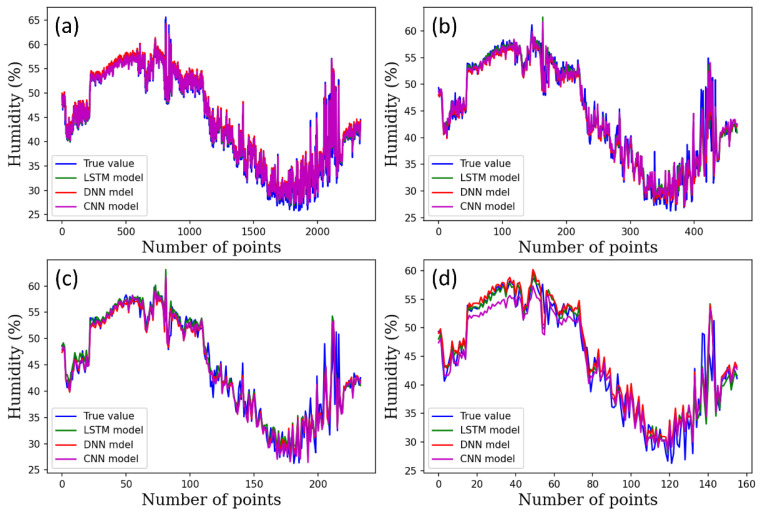
Humidity prediction results using DL models per time steps, (**a**) 1-min time step, (**b**) 5-min time steps, (**c**) 10-min time steps, (**d**) 15-min time steps.

**Figure 6 plants-12-02316-f006:**
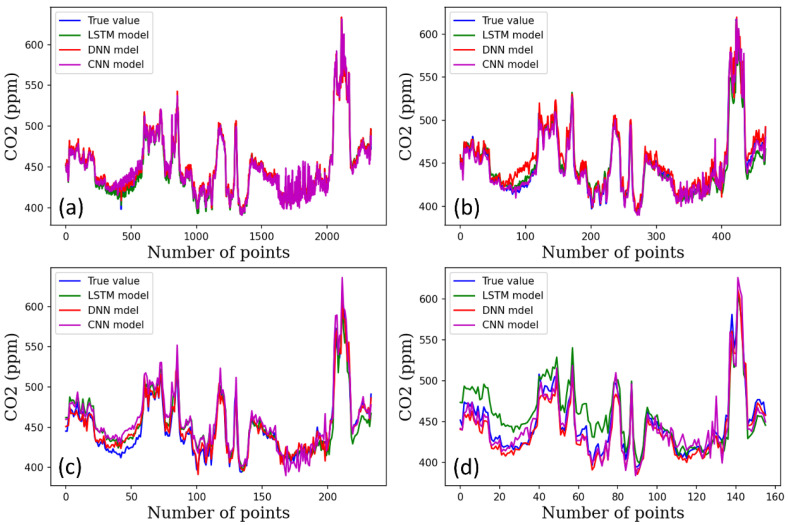
CO_2_ prediction results using DL models per time steps, (**a**) 1-min time step, (**b**) 5-min time steps, (**c**) 10-min time steps, (**d**) 15-min time steps.

**Figure 7 plants-12-02316-f007:**
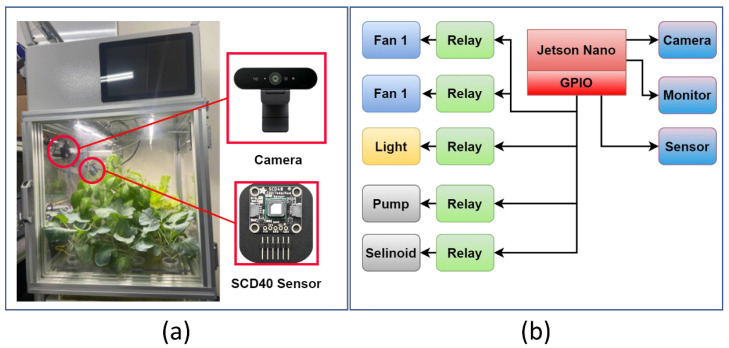
Experimental setup, (**a**) a prototype hydroponic greenhouse, (**b**) scheme of condition monitoring of the greenhouse.

**Figure 8 plants-12-02316-f008:**
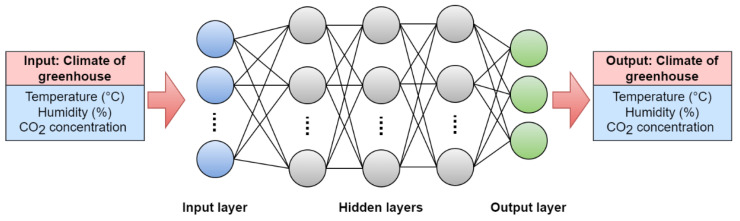
DNN architecture was used in this study.

**Figure 9 plants-12-02316-f009:**
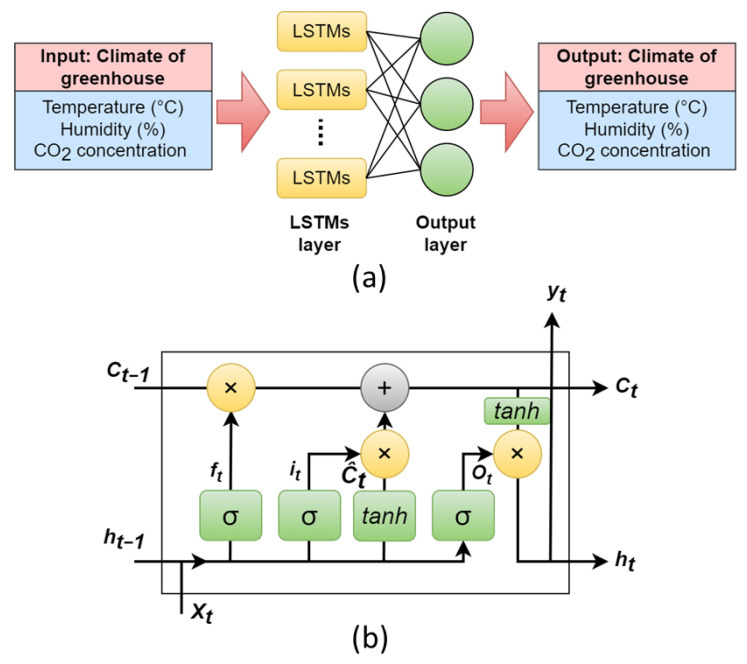
(**a**) Architecture of the LSTM model used in this study and (**b**) the structure of a single cell of LSTM.

**Figure 10 plants-12-02316-f010:**
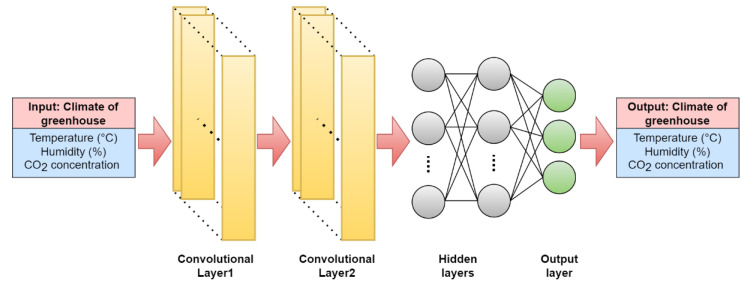
1D-CNN architecture used in this study.

**Table 1 plants-12-02316-t001:** Comparison of the DNN model performance at different time intervals.

	DNN-1			DNN-5		
	Temperature	Humidity	CO_2_	Temperature	Humidity	CO_2_
R^2^	0.98	0.96	0.81	0.99	0.95	0.97
MAE	0.005	0.02	0.01	0.004	0.02	0.003
RMSE	0.006	0.03	0.01	0.005	0.03	0.004
SEP (%)	0.6	3.04	3.45	0.4	3.65	1.08
	**DNN-10**			**DNN-15**		
	**Temperature**	**Humidity**	**CO_2_**	**Temperature**	**Humidity**	**CO_2_**
R^2^	0.95	0.94	0.81	0.98	0.93	0.91
MAE	0.008	0.023	0.009	0.005	0.024	0.006
RMSE	0.009	0.031	0.011	0.006	0.035	0.007
SEP (%)	0.88	3.8	2.99	0.56	4.22	2.15

**Table 2 plants-12-02316-t002:** Comparison of the LSTM model performance at different time intervals.

	LSTM-1			LSTM-5		
	Temperature	Humidity	CO_2_	Temperature	Humidity	CO_2_
R^2^	0.99	0.96	0.96	0.99	0.94	0.93
MAE	0.004	0.018	0.004	0.004	0.02	0.005
RMSE	0.004	0.027	0.0045	0.005	0.032	0.006
SEP (%)	0.4	3.09	1.35	0.46	3.54	1.77
	**LSTM-10**			**LSTM-15**		
	**Temperature**	**Humidity**	**CO_2_**	**Temperature**	**Humidity**	**CO_2_**
R^2^	0.95	0.94	0.89	0.96	0.93	0.81
MAE	0.008	0.022	0.007	0.007	0.024	0.009
RMSE	0.01	0.034	0.008	0.008	0.037	0.01
SEP (%)	0.94	3.85	2.38	0.84	4.18	3.17

**Table 3 plants-12-02316-t003:** Comparison of the 1D-CNN model performance at different time intervals.

	1D-CNN-1			1D-CNN-5		
	Temperature	Humidity	CO_2_	Temperature	Humidity	CO_2_
R^2^	0.96	0.96	0.97	0.95	0.93	0.92
MAE	0.007	0.02	0.003	0.006	0.025	0.005
RMSE	0.008	0.03	0.004	0.009	0.035	0.007
SEP (%)	0.79	3.46	1.1	0.69	4.2	1.84
	**1D-CNN-10**			**1D-CNN-15**		
	**Temperature**	**Humidity**	**CO_2_**	**Temperature**	**Humidity**	**CO_2_**
R^2^	0.97	0.95	0.74	0.88	0.94	0.74
MAE	0.005	0.02	0.009	0.013	0.024	0.011
RMSE	0.008	0.03	0.012	0.015	0.034	0.012
SEP (%)	0.56	3.48	3.02	1.45	4.21	3.95

## Data Availability

The data presented in this study are available on request from the corresponding author.

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
