# Peer review of "Performance Analysis of Time Series Deep Learning Models for Climate Prediction in Indoor Hydroponic Greenhouses at Different Time Intervals"

_plants, 2023, doi:10.3390/plants12122316_

Round 1
Reviewer 1 Report
The authors predict three variables (temperature, humidity, and C02) at horizons of 1, 5, 10, and 15 minutes. The data is collected for a week from a small (50x50x60 cm) hydroponic greenhouse. Three forecasting algorithms are tested: a 128x64x32 neural network, an LSTM, and a 1D convolutional neural network. Four performance metrics are calculated (R^2, MAE, RMSE, and %SEP). Overall, the LSTM performs best. The methods are clearly explained, and the presentation of the results in comprehensive.
More motivation for why this is an interesting problem is needed. How would the predicted variables be used in a control scheme? How do these variables effect crop production?
In the greenhouse, how are the fans, lights, and pumps controlled? Is there a control strategy that can better explain the predicted values?
What are the input features for the feedforward neural network?
Line 139: What is meant by “DL models are also trained on the same hardware”? Are the models trained on the Jetson Nano? I doubt this. Or do you mean that the models were trained on the data collected by the hardware? I think this is more likely.
Line 161: Here you state that there are two hidden layers, but figure 2 shows three hidden layers.
Line 165: You state that you use the Levenberg–Marquardt training algorithm, but on line 170 you state that you use the Adam optimizer. Are you using both algorithms? If so, how have you combined them?
Minor editing is all that is needed.
Author Response
Comments and Suggestions for Authors
Reviewer #1
The authors predict three variables (temperature, humidity, and C02) at horizons of 1, 5, 10, and 15 minutes. The data is collected for a week from a small (50x50x60 cm) hydroponic greenhouse. Three forecasting algorithms are tested: a 128x64x32 neural network, an LSTM, and a 1D convolutional neural network. Four performance metrics are calculated (R^2, MAE, RMSE, and %SEP). Overall, the LSTM performs best. The methods are clearly explained, and the presentation of the results in comprehensive.
First, we would like to thank you for devoting precious time to reading our paper and giving comments.
Reviewer#1, Concern # 1: More motivation for why this is an interesting problem is needed. How would the predicted variables be used in a control scheme? How do these variables effect crop production?
Author response: The predicted variables can be utilized in a control scheme to dynamically adjust and regulate various environmental factors within the greenhouse. For instance, based on the predicted temperature, the control system can activate or deactivate heating or cooling systems to maintain the optimal temperature range for crop growth. Similarly, the predicted humidity levels can guide the activation of humidifiers or dehumidifiers to ensure the appropriate moisture content in the air. Additionally, the predicted CO2 levels can be used to regulate ventilation systems and control the inflow and outflow of air to maintain the desired CO2 concentration. By integrating the predicted variables into a control scheme, we can create an automated and optimized environment for crop production. This has direct implications on crop growth, as optimal climate conditions are vital for achieving higher yields, reducing disease incidence, and improving overall crop quality. Precise control of temperature, humidity, and CO2 levels can influence various physiological processes in plants, including photosynthesis, transpiration, and nutrient uptake. By maintaining these variables within the optimal range, we can promote healthy plant growth, improve resource utilization, and ultimately enhance crop productivity.
Reviewer#1, Concern # 2: In the greenhouse, how are the fans, lights, and pumps controlled? Is there a control strategy that can better explain the predicted values?
Author response: In the greenhouse, the control of fans, lights, and pumps is an essential aspect of creating an optimal environment for crop growth. These components are typically controlled through an automated system that takes into account various factors, including the predicted values of climate variables. The control strategy for managing fans, lights, and pumps can be based on a combination of real-time sensor data and the predicted values of temperature, humidity, and CO2 levels. The goal is to maintain the desired setpoints for these variables and ensure their stability within the optimal range. For example, fans can be controlled to regulate air circulation and maintain proper ventilation within the greenhouse. Based on the predicted temperature and humidity levels, the control system can adjust the fan speed or activate/deactivate specific fans to achieve the desired airflow and prevent overheating or excessive moisture buildup. Similarly, the control of lights in the greenhouse can be guided by the predicted values of climate variables. The lighting duration and intensity can be adjusted to provide the optimal light spectrum and duration for different stages of plant growth. By considering the predicted values, the control system can ensure that the lighting conditions are tailored to the specific requirements of the crops. Pumps are used in hydroponic systems to deliver nutrient solutions to the plants. The control of pumps can be synchronized with the predicted climate variables to optimize the delivery of nutrients. For instance, based on the predicted water consumption by the plants (indicated by the humidity levels), the control system can adjust the pump operation to deliver the appropriate amount of nutrient solution, ensuring efficient water and nutrient management.
As mentioned in the manuscript from line 131-134, This study focuses on only greenhouse environmental prediction aspects which is a part of our main research (Integration of AI Predictions and IoT Technologies for Autonomous Crop Production in Greenhouses). Therefore, we will cover all AI predictions and control system and IoT technologies in further research work.
Reviewer#1, Concern # 3: What are the input features for the feedforward neural network?
Author response: As we mentioned in the manuscript, Input features for all DL models are the same. Temperature, Humidity and CO2 are as input features for DL models.
Reviewer#1, Concern # 4: Line 139: What is meant by “DL models are also trained on the same hardware”? Are the models trained on the Jetson Nano? I doubt this. Or do you mean that the models were trained on the data collected by the hardware? I think this is more likely.
Author response: Apologies for any confusion caused by the statement. When it is mentioned that "DL models are also trained on the same hardware," it refers to the training process being conducted using the Jetson Nano device itself. Typically, during the training phase of deep learning models, the training process requires significant computational resources, including memory and processing power. While Jetson Nano is capable of training small to moderate-sized models, it may have limitations when it comes to training larger and more complex models due to its lower computational capacity compared to high-end GPUs or dedicated training servers. Therefore, it is important to consider the size and complexity of the models being trained and assess whether Jetson Nano is capable of handling the training process efficiently. So, we decided to train all DL models on the Jetson Nano hardware because of the low complexity of the models and relatively small size of dataset.
Reviewer#1, Concern # 5: Line 161: Here you state that there are two hidden layers, but figure 2 shows three hidden layers.
Author response: According to the comment of the reviewer, we have updated the sentence. It should be 3 hidden layers in the text also. This is a typo error.
Reviewer#1, Concern # 6: Line 165: You state that you use the Levenberg–Marquardt training algorithm, but on line 170 you state that you use the Adam optimizer. Are you using both algorithms? If so, how have you combined them?
Author response: According to the comment of the reviewer, we have updated the sentence. It should be Adam optimization instead of Levenberg-Marquardt. This is a typo error.

Reviewer 2 Report

The English language used in the scientific article seems to be precise and accurate.
Author Response
Comments and Suggestions for Authors
Reviewer #2
Reviewer#2, Concern # 1: After carefully reviewing your work, I would like to provide you with some feedback and suggestions for further improvement.
Firstly, we appreciate your study on the evaluation of three deep learning models (Deep Neural Network, LSTM, and 1D Convolutional Neural Network) for climate prediction in indoor hydroponic greenhouses. This topic is of great significance, given the increasing popularity of these greenhouses for sustainable food production.
The comparative analysis of the models at different time intervals, using a dataset collected over one week, is commendable. The results demonstrate that all models perform well in predicting temperature, humidity, and CO2 concentration within the greenhouse environment. It is particularly interesting to note that the LSTM model outperforms the others at shorter time intervals, highlighting the importance of selecting the appropriate time interval for accurate predictions.
However, we would like to address some limitations in your study. Firstly, the short duration of data collection over one week may limit the generalizability of your findings. We encourage you to consider extending the data collection period to capture a wider range of climatic variations and potential outliers.
Additionally, it would be valuable to include a comparison with alternative methods for climate prediction in indoor hydroponic greenhouses. This would provide a more comprehensive understanding of the effectiveness of the deep learning models in comparison to other approaches.
We also recommend conducting further testing and validation, considering external factors that may influence climate predictions. Factors such as changes in external environmental conditions or specific elements within the hydroponic cultivation system should be considered for a more robust analysis.
Overall, we recognize the importance of your study in advancing smart control systems in hydroponic greenhouses and promoting sustainable food production. Your insights provide a valuable contribution to the scientific community. We kindly request that you address the points to enhance the quality and impact of your research.
Author response: Thank you for providing the reviewer's comments. We appreciate the valuable feedback and would like to address the limitations mentioned in our study.
Regarding the short duration of data collection, we acknowledge that a one-week period may limit the generalizability of our findings. We understand the importance of capturing a wider range of climatic variations and potential outliers. In light of this, we plan to extend the data collection period in future studies to obtain a more comprehensive dataset that encompasses a broader range of environmental conditions. This will enhance the reliability and generalizability of our findings.
We also acknowledge the suggestion to include a comparison with alternative methods for climate prediction in indoor hydroponic greenhouses. We agree that such a comparison would provide a more comprehensive understanding of the effectiveness of deep learning models in comparison to other approaches. In our future work, we will include a comparative analysis with alternative methods to evaluate their performance and provide a more robust assessment of the deep learning models' efficacy.
Furthermore, we appreciate the recommendation to conduct further testing and validation, considering external factors that may influence climate predictions. We recognize the importance of accounting for external environmental conditions and specific elements within the hydroponic cultivation system. In our future research, we will incorporate these factors into our analysis to ensure a more thorough and comprehensive evaluation of the deep learning models' predictive capabilities.
Thank you once again for your valuable feedback. We will address these limitations in our future work to enhance the robustness and applicability of our study.

Round 2
Reviewer 1 Report
In their response letter, the authors have done an excellent job addressing the comments I made. However, they have not incorporated these responses into their manuscript. The questions I have raised are likely to also be raised by future readers. Thus, the authors need to incorporate their responses into the manuscript. Doing so will make the paper stronger.
Line 139: To make it even clearer that the models were trained on the Jetson Nano, change “same hardware” to “Jetson Nano”. Also incorporate your response to Concern #4 into your manuscript. It will make it a stronger paper.
Author Response
Once again, we would like to thank you for devoting precious time to reading our paper and giving comments.
Reviewer#1, Concern # 1: In their response letter, the authors have done an excellent job addressing the comments I made. However, they have not incorporated these responses into their manuscript. The questions I have raised are likely to also be raised by future readers. Thus, the authors need to incorporate their responses into the manuscript. Doing so will make the paper stronger.
Author response: According to the reviewer’s comment, we have incorporated our previous responses into the manuscript from Line 134-149 and from Line 150-171.
Reviewer#1, Concern # 2: Line 139: To make it even clearer that the models were trained on the Jetson Nano, change “same hardware” to “Jetson Nano”. Also incorporate your response to Concern #4 into your manuscript. It will make it a stronger paper.
Author response: According to the reviewer’s comment, we have changed the phrase “same hardware” to “Jetson Nano” Line 174.

Reviewer 2 Report
The article has fulfilled the requirements.
Author Response
We would like to thank you for devoting precious time to give review for our paper.
Best regards.
Authors
Round 3
Reviewer 1 Report
The authors have addressed all my concerns.